# A Family of Fitness Landscapes Modeled through Gene Regulatory Networks

**DOI:** 10.3390/e24050622

**Published:** 2022-04-29

**Authors:** Chia-Hung Yang, Samuel V. Scarpino

**Affiliations:** 1Network Science Institute, Northeastern University, Boston, MA 02115, USA; 2Physics Department, Northeastern University, Boston, MA 02115, USA; 3Roux Institute, Northeastern University, Boston, MA 02115, USA; 4Institute for Experiential AI, Northeastern University, Boston, MA 02115, USA; 5Santa Fe Institute, Santa Fe, NM 87501, USA; 6Vermont Complex Systems Center, University of Vermont, Burlington, VT 05405, USA

**Keywords:** fitness landscapes, gene regulatory networks, coarse-graining, biological computation, graph theory

## Abstract

Fitness landscapes are a powerful metaphor for understanding the evolution of biological systems. These landscapes describe how genotypes are connected to each other through mutation and related through fitness. Empirical studies of fitness landscapes have increasingly revealed conserved topographical features across diverse taxa, e.g., the accessibility of genotypes and “ruggedness”. As a result, theoretical studies are needed to investigate how evolution proceeds on fitness landscapes with such conserved features. Here, we develop and study a model of evolution on fitness landscapes using the lens of Gene Regulatory Networks (GRNs), where the regulatory products are computed from multiple genes and collectively treated as phenotypes. With the assumption that regulation is a binary process, we prove the existence of empirically observed, topographical features such as accessibility and connectivity. We further show that these results hold across arbitrary fitness functions and that a trade-off between accessibility and ruggedness need not exist. Then, using graph theory and a coarse-graining approach, we deduce a mesoscopic structure underlying GRN fitness landscapes where the information necessary to predict a population’s evolutionary trajectory is retained with minimal complexity. Using this coarse-graining, we develop a bottom-up algorithm to construct such mesoscopic backbones, which does not require computing the genotype network and is therefore far more efficient than brute-force approaches. Altogether, this work provides mathematical results of high-dimensional fitness landscapes and a path toward connecting theory to empirical studies.

## 1. Introduction

Since its introduction by Wright [1], the concept of fitness landscapes has grown and matured into a cornerstone of biology [2,3,4]. A fitness landscape consists of a space of genotypes that are mutually accessible through mutations and a fitness value associated with the phenotype each genotype encodes. In this context, fitness describes the evolutionary potential of each genotype, and the set of navigable genotypes on these landscapes is termed the genotype network [5]. Continuing with this metaphor, the evolution of a population can be depicted as a trajectory wandering on the fitness landscape. As a consequence, the topography of a fitness landscape sheds light on various evolutionary processes, including constraints on adaptation [6,7,8,9], speciation via genetic incompatibilities [10,11], (dis)advantages of sexual reproduction and recombination [12,13,14], the repeatability/reversibility (or not) of evolutionary trajectories [15,16,17,18], and the role of neutral networks—components of the genotype network with the same fitness—in epochal evolution [19,20,21,22,23,24].

Despite being introduced by Wright [1], Fisher’s 1930 geometric model of adaptation is the first mathematical model of evolution on what we now call fitness landscapes [15,25,26]. Later work by Kingman [27] and Kauffman and Levin [28] constructed what they termed a “house of cards” (HoC) model where fitness values for each genotype are drawn independently from a specified probability distribution. Building on the HoC model, Kauffman and Weinberger [29] introduced the NK model, which forces each locus to interact with a fixed number of other loci and where a genotype’s fitness becomes the sum of the fitness contributions of every interaction group. More recently, the “rough Mount Fuji model” [30,31] combines the HoC landscape with an additional field penalizing a genotype’s Hamming distance away from a referenced genotype with the optimal fitness. The dependence of a genotype’s fitness on that of neighboring genotypes is thought to be a key feature of empirical fitness landscapes.

Over the past three decades, the fitness landscapes for various organisms, including bacteria [32,33,34], fungi [35,36], and fruit flies [37], have been empirically reconstructed. While the number of genotypes included in these early landscapes was limited, modern sequencing techniques and high-throughput analyses have enabled the construction of many large landscapes. Notable studies have been conducted in HIV [38,39], yeast [40], *E. coli* [41], jellyfish [42], human cancers [43], human stem cells [44], and DNA/RNA networks [45,46,47,48,49]. Comprehensive landscapes for multiple eukaryotic species have also been analyzed based on the binding affinity of transcription factors [50] and after accounting for the ecological context the species experiences [51]. What emerged from these studies is a set of prominent topographical features conserved across diverged taxa [52].

Together, empirical and modeled fitness landscapes exhibit three key topographical features. First, fitness landscapes are more often “rugged” than smooth [52]. The degree of ruggedness can be assessed via a variety of measures, such as the roughness of the slope ratio [53,54] and the number of local fitness maxima [55], which are often strongly correlated with each other [4]. Empirical studies typically show moderate ruggedness in the observed fitness landscapes [32,33,34,36,50,56]. The degree of ruggedness in these empirical landscapes is less than the HoC model assumes and comparable to a fine-tuned NK model or rough Mount Fuji model [4]. Second, fitness landscapes reveal mutational trajectories from one genotype to another where the fitness is non-decreasing, which implies accessibility (typically to a fitness optimum) across the landscape [32,57,58,59,60]. Lastly, whereas the inaccessible region in the HoC model expands when distant from the fitness optimum [61], other models find accessible trajectories despite high genotypic dimensionality [62,63].

Due to the often pervasive interaction between loci, determining phenotype from genotype can have a high degree of computational complexity [64,65]. Many existing fitness landscape models have dealt with this complexity by strongly constraining the state-space of possible genotypic interactions and/or reducing the complexity of how information is processed when mapping genotype to phenotype. For example, studies have focused on the folded structure of short RNA sequences, where the resulting stability or affinity is a fitness proxy [66,67,68,69,70], networks of molecular/genetic pathways whose expression pattern or homeostasis determines fitness [71,72,73,74,75], and modular mutational effects at different loci in Fisher’s geometric model [76].

Here, we model the genotype–phenotype map using the pathway framework of gene regulatory networks (GRNs), where mechanistic knowledge of how phenotypes are computed from genotypes is encoded in the GRN (see [77,78] for a more formal introduction). To study the fitness landscapes induced by GRN evolution, we integrate the pathway framework into a family of fitness landscape models where the fitness value is uniquely determined by the phenotype corresponding to the regulatory outcome of a genotype. For a fitness landscape of GRNs, we first prove the existence of two key topographical features: (a) GRNs with the same phenotype are themselves connected in the underlying genotypic network, and (b) there exists accessible trajectories between all pairs of GRNs with similar phenotypes. Second, utilizing the idea of symmetries and automorphisms in the genotype network, we coarse-grain GRNs into groups with equivalent roles in the fitness landscapes and deduce an underlying mesoscopic structure with which we can predict the trajectory of evolution with minimal complexity. Lastly, using this coarse-graining, we develop a bottom-up algorithm for constructing the underlying fitness landscape of GRNs, which does not require computing the genotype network and is thus more efficient than the conventional brute-force approach.

## 2. Methods

Here, we introduce a family of fitness landscape models where the genotype-phenotype mapping is constructed from regulatory interactions. We first summarize a modeling framework of GRNs proposed in our previous work [77,78], termed the pathway framework, and then fitness landscape models of GRNs built upon the pathway framework.

### 2.1. Pathway Framework of GRNs

Genotypes in the pathway framework of GRNs contain all necessary information to construct a regulatory network [77,78]. More specifically, alleles at each locus include both a transcription activator and a protein product, which means the regulatory interactions among the loci can be deduced by connecting genes whose expression product corresponds to the activator of another. Compared to existing work on regulatory circuits—where mutations are modeled as rewiring a single interaction between genes [79,80]—the pathway framework considers a mutation as changing the activator/product of a gene. Lastly, the phenotype is determined by the set of loci reached in a regulatory cascade induced by external stimuli. These stimuli could be completely external to the individual or simply come from another regulatory network in the organism. For additional details on the pathway framework, see [77,78] and Figure 1 for illustration.

In this work, when building the family of fitness landscape models, we restrict the pathway framework with four assumptions. First, we consider a fixed set of genes underlying the genotypes, i.e., gene duplication and deletion events are excluded. Second, we assume a fixed underlying collection of proteins that can possibly exist in the organism. Third, we consider the case where a gene’s expression is activated by a specific protein, and it generates only one protein product. Fourth, we assume that the associated chemical state of each protein is modeled as a Boolean/binary variable (present or absent), and external environmental signals stimulate the existence of specific proteins in the organism. As a consequence, the Boolean state of a phenotype-related protein is determined by whether it is reached by a regulatory cascade starting from an initial stimulus.

While the above assumptions seem naive, as we will show in Section 3, this simplified model still predicts the topographical features observed in empirical landscapes (see Section 1). As a result of these assumptions, we are able to derive rigorous theoretical insights into GRN evolution and obtain fitness landscapes consistent with far more complicated models. We believe these assumptions are conservative with respect to the biology and a justified starting point for modeling fitness landscapes, and we discuss the implications of these assumptions and possible extensions to the model in Section 4.

### 2.2. Fitness Landscape of GRNs under the Pathway Framework

Let Γ and Ω be the fixed, underlying collection of loci and proteins, respectively. A genotype is represented by its GRN *g* such that every locus γ∈Γ is associated with a protein activator/product pair $e_{g}\left(\gamma \right)=\left(u,v\right)$, u,v∈Ω. Equivalently, any GRN is a directed graph with |Ω| nodes labeled by the proteins Ω and |Γ| edges labeled by the loci Γ. In the rest of this paper, we will use the terminology “source/target” node of edge γ interchangeably to refer to the protein activator/product of locus γ. We also write G to be the set of all GRNs with the underlying loci Γ and proteins Ω.

The backbone of a fitness landscape of GRNs, i.e., the genotype network, is an undirected network of networks encoding the mutational relationship between the GRNs. Let *G* be the genotype network, and we denote its mega-nodes by V(G)=G and its edges by E(G). There is an edge $\left(g_{1},g_{2}\right)\in E\left(G\right)$ between any GRNs $g_{1},g_{2}\in G$ when they only differ by the allele of a single locus γ, $e_{g_{1}}\left(\gamma \right)\neq e_{g_{2}}\left(\gamma \right)$. In other words, $g_{1}$ and $g_{2}$ are connected in *G* when they can be transformed into each other through one edge rewiring.

Furthermore, we write $x_{\omega }$ to be the binary state of protein ω∈Ω, where $x_{\omega }=1$ indicates the presence of ω, and $x_{\omega }=0$ designates its absence. We also partition Ω into three disjoint groups: (a) proteins $\Omega_{0}$ whose presence is externally stimulated by the given environment, (b) proteins $\hat{\Omega }$ whose states influence the fitness value, and (c) the remaining ones, which we call the dummy proteins $\Omega^{\prime }$ since their specific identities are irrelevant to the external environment and the resultant phenotypes/fitness. (In this paper, we assume that the stimuli $\Omega_{0}$ must be proteins that cannot be produced by expression, and we leave no constraint to the fitness-relevant and dummy proteins $\hat{\Omega }$ and $\Omega^{\prime }$).

A phenotype is then treated as a vector of zeros and ones, where each entry corresponds to the binary state of a protein in $\hat{\Omega }$. The resultant phenotype $x_{\hat{\Omega }}\left(g\right)$ of a GRN *g* is determined by the reachability in *g*: For any $\omega \in \hat{\Omega }$, $x_{\omega }=1$ if and only if there is a stimulus $\omega_{0}\in \Omega_{0}$ and a path from $\omega_{0}$ to ω in *g*, which represents a chain of sequentially expressed genes that generates protein ω. Finally, the fitness *f* is simply a function of the phenotype $x_{\hat{\Omega }}\left(g\right)$.

Combined, a fitness landscape of GRNs is characterized by three key elements: the genotype network *G*, the external stimuli $\Omega_{0}$, and the fitness function of phenotype *f* (which implicitly identifies the fitness-relevant proteins $\hat{\Omega }$). The genotype network *G* serves as the skeleton of the fitness landscape, whereas the environment-dependent stimuli $\Omega_{0}$ and fitness function *f* determine the phenotypes of GRNs and their selective advantages.

## 3. Results

In this work, we derive three theoretical insights into fitness landscape models using GRNs as the embedded genotype–phenotype mapping. First, we show that the resulting family of fitness landscapes must always contain two topographical properties: connectivity, i.e., GRNs with the same phenotype can be mutually reached via mutations, and accessibility, i.e., that any GRN can be reached from an arbitrary less-fit GRN (once certain similarity criterion is met). Second, we propose a mesoscopic coarse-graining for fitness landscapes, which is a more compact alternative to analyzing evolutionary processes than the original landscape. This mesoscopic backbone recognizes “symmetries” in the genotype network, and it aggregates GRNs with the same role in the fitness landscape into a single representative genotype. Third, we provide a bottom-up approach to algorithmically construct this mesoscopic backbone and demonstrate its efficiency over coarse-graining the genotype network using brute force.

### 3.1. Connectivity and Accessibility in a Fitness Landscape of GRNs

A fitness landscape model of GRNs features a handful of properties that have either been discovered in empirical fitness landscapes or investigated mathematically. First, its underlying space, i.e., the genotype network *G*, presents immense dimensionality. Second, the fitness function *f* is flexible and can effectively tune the ruggedness of the fitness landscape. For example, a highly rugged “holely” landscape can be modeled by a binary *f* such that any GRN g∈G has high fitness once some single protein $\omega \in \hat{\Omega }$ is present, $x_{\omega }=1$, and otherwise, *g* has low/zero fitness. Because one can always find several mutational neighbors of *g* whose phenotype shows an opposite state $x_{\omega }$, the resultant fitness landscape is inevitably rugged. In what follows, we further show that fitness landscape models of GRNs must hold the characteristics of connectivity and accessibility.

Let y be a phenotype and denote by $G_{y}$ the set of all GRNs with phenotype y, i.e., $x_{\hat{\Omega }}\left(g\right)=y$ for $g\in G_{y}$, under the given external stimuli $\Omega_{0}$. We also write $\Omega_{y}^{+}$ to be the required-present proteins in the phenotype y, so $x_{\omega }=1$ for $\omega \in \Omega_{y}^{+}$ and $x_{\omega^{\prime }}=0$ for any other $\omega^{\prime }\in \hat{\Omega }⧵\Omega_{y}^{+}$. Note that the number of required-present proteins $|\Omega_{y}^{+}|$ is bounded from above by the number of loci |Γ| since any present protein that is not a stimulus must be triggered by the expression of some locus.

We observe that some GRNs ${\tilde{G}}_{y}\subset G_{y}$ play a “central” role among GRNs with the same phenotype y. Specifically, for any $\tilde{g}\in {\tilde{G}}_{y}$, all the edges in $\tilde{g}$ point from the stimuli $\Omega_{0}$ to the required-present proteins $\Omega_{y}^{+}$, and each $\omega \in \Omega_{y}^{+}$ is targeted by at least one edge in $\tilde{g}$. We demonstrate an example of such $\tilde{g}$ in Figure 2a. These ${\tilde{G}}_{y}$ are deemed central because they can be reached by any GRN $g\in G_{y}$ through mutations among $G_{y}$ themselves: First, for every edge in *g* that points to an $\omega \in \Omega_{y}^{+}$, we rewire the edge such that it still points to ω but now from an $\omega_{0}\in \Omega_{0}$. Arbitrarily rewiring the remaining edges between $\Omega_{0}$ and $\Omega_{y}^{+}$ then leads to some central GRN in ${\tilde{G}}_{y}$ (see Figure 2a).

In addition, if the phenotype y has strictly less required proteins than the number of loci, the central GRNs $G_{y}$ are mutually reachable by edge rewiring among ${\tilde{G}}_{y}$. There is always a redundant edge whose rewiring makes no change to the phenotype, and it helps us rewire each edge to any desired source/target pair between $\Omega_{0}$ and $\Omega_{y}^{+}$ (see Figure 2b), which subsequently creates a chain of mutations between any $\tilde{g},{\tilde{g}}^{\prime }\in {\tilde{G}}_{y}$. These results implicate that, for any phenotype y with $|\Omega_{y}^{+}\left|<|\Gamma \right|$ and any $g_{1},g_{2}\in G_{y}$, there is always a mutational trajectory between $g_{1}$ and $g_{2}$ that only traverses over GRNs in $G_{y}$, especially through the central ones ${\tilde{G}}_{y}$ (see Figure 2c). In the extreme case where $|\Omega_{y}^{+}\left|=|\Gamma \right|$, however, $G_{y}$ fragments into multiple connected components (detailed in Appendix A).

Next, we turn to accessibility between GRNs of different phenotypes y and $y^{\prime }$, where without loss of generality $f\left(y^{\prime }\right)\geq f\left(y\right)$. We observe that, if $|\Omega_{y}^{+}\cup \Omega_{y^{\prime }}^{+}|\;\leq \;|\Gamma |+1$, there are always two “peripheral” GRNs $\hat{g}\in G_{y}$ and ${\hat{g}}^{\prime }\in G_{y^{\prime }}$, which only differ by one edge rewiring. To be more specific, there are two independent chains in $\hat{g}$, one of which begins with a stimulus $\omega_{0}\in \Omega_{0}$ and sequentially connects the proteins required to be present in y but not in $y^{\prime }$, i.e., $\Omega_{y}^{+}⧵\Omega_{y^{\prime }}^{+}$, while the other consecutively joins $\Omega_{y^{\prime }}^{+}⧵\Omega_{y}^{+}$. The rest of the edges in $\hat{g}$ merely point from $\Omega_{0}$ to $\Omega_{y}^{+}\cap \Omega_{y^{\prime }}^{+}$, and each $\omega \in \Omega_{y}^{+}\cap \Omega_{y^{\prime }}^{+}$ is targeted by at least one edge (see example in Figure 3, left). The other GRN ${\hat{g}}^{\prime }$ only differ from *g* by the first edge in the chain of $\Omega_{y}^{+}⧵\Omega_{y^{\prime }}^{+}$, which is rewired such that it points from the stimulus $\omega_{0}$ to the first node in the chain of $\Omega_{y^{\prime }}^{+}⧵\Omega_{y}^{+}$ (Figure 3, right).

Our observation suggests that there is a sequence of mutations with non-decreasing fitness from any GRN $g\in G_{y}$ to any GRN $g^{\prime }\in G_{y^{\prime }}$, as long as $|\Omega_{y}^{+}\cup \Omega_{y^{\prime }}^{+}|\leq |\Gamma |+1$. In particular, when $|\Omega_{y}^{+}\left|<|\Gamma \right|$, the mutational trajectory starting at *g* first traverses within $G_{y}$ to a peripheral GRN and then transitions into $G_{y^{\prime }}$ to reach $g^{\prime }$. An analogous trajectory exists even under the extreme scenario $|\Omega_{y}^{+}\left|=|\Gamma \right|$ (see Appendix A). We also note that if the number of fitness-relevant proteins is $|\hat{\Omega }|\leq |\Gamma |+1$, then the condition $|\Omega_{y}^{+}\cup \Omega_{y^{\prime }}^{+}|\leq |\Gamma |+1$ is assuredly satisfied for any two phenotypes y and $y^{\prime }$. As a corollary, if $|\hat{\Omega }|\leq |\Gamma |+1$, the fitness optimum will always be accessible.

### 3.2. Mesoscopic Skeleton Derived from “Symmetries” in the Genotype Network of GRNs

Because the number of possible GRNs grows super-exponentially as the underlying loci and proteins expand, constructing the genotype network becomes extremely challenging beyond a small Γ and Ω. Here, we present a more compact skeleton of the fitness landscape of GRNs based on “symmetries” in the genotype network.

As the underlying space of a fitness landscape of GRNs, the genotype network *G* appears to contain redundant information. On the one hand, GRNs leading to the identical phenotype are deemed to have equal fitness. On the other hand, given any GRN, for example, the mega-node rounded by orange in Figure 4a, one can always find some other GRN such that their neighborhoods in *G* are locally similar, e.g., the mega-node rounded by blue. This simple demonstration suggests that the structure of the genotype network *G* is not arbitrary; instead, some structural symmetries exist.

In graph theory, symmetries in a network are formally described through the network’s automorphisms. An automorphism of a graph is a way to shuffle the labels of its nodes such that the graph remains identical before and after shuffling. For instance, in Figure 5b, exchanging nodes 2 and 3 generates the same network and is thus an automorphism, whereas exchanging nodes 2 and 4 is not because there is an edge from 2 to 3 after shuffling. Formally, an automorphism of the genotype network *G* is a permutation σ of all plausible GRNs G=V(G) such that, for any $g_{1},g_{2}\in G$, $\left(\sigma \left(g_{1}\right),\sigma \left(g_{2}\right)\right)\in E\left(G\right)$ if and only if we also have $\left(g_{1},g_{2}\right)\in E\left(G\right)$. (A permutation of G is a mapping σ:G→G where no two GRNs are mapped to the same GRN, i.e., $\sigma \left(g_{1}\right)\neq \sigma \left(g_{2}\right)$ if $g_{1}\neq g_{2}$ for any $g_{1},g_{2}\in G$.) Once two GRNs *g* and $g^{\prime }$ are related through an automorphism σ of *G*, e.g., $g^{\prime }=\sigma \left(g\right)$, they share the same mega-node properties that are fully determined by the connections in the genotype network (see Proposition A1).

Furthermore, automorphisms partition the GRNs by their roles in the genotype network through the mathematical concept of equivalence classes. For a high-level and general description, imagine a set of elements and a group of operations acting on them. Each operation turns one element into another, and these two elements are related by the operation, which describes the similarity between them. An equivalence class consists of elements that are mutually related by any operation, and the set of elements is said to be partitioned into equivalence classes under the action of the operations (see Figure 5a for an illustrative example). For automorphisms $\Sigma (G^{\prime })$ of a graph $G^{\prime }$, the equivalence classes of nodes $V(G^{\prime })$ under the action of $\Sigma (G^{\prime })$ then gather nodes with a similar “structural position” in $G^{\prime }$ (Figure 5c).

However, to reveal GRNs with identical roles in a fitness landscape, these automorphisms also need to preserve the phenotype. Denote by $\Sigma_{x}\left(G\right)$ the set of such automorphisms of *G*, i.e., for any $\sigma \in \Sigma_{x}\left(G\right)$, and GRN g∈G, σ(g) and *g* have the same phenotype. The equivalence classes of mega-nodes V(G) under the action of phenotype-preserving automorphisms $\Sigma_{x}\left(G\right)$ then unite GRNs that (a) show similar mutational relationships with others and (b) lead to the same fitness due to their identical phenotype. We will mildly abuse the terminology to call them the *equivalence classes of GRNs*, which we denote by Θ, and each θ∈Θ is a set of GRNs related through $\Sigma_{x}\left(G\right)$. Crucially, since the mutational relationship and the resultant phenotype are the two components that characterize a GRN in the fitness landscape, GRNs in a θ∈Θ are deemed equivalent semantically, and they can be reduced to an arbitrary representative among them. Therefore, the equivalence classes of GRNs provide an efficient way to depict the underlying space of the fitness landscape.

However, what exactly composes the phenotype-preserving automorphisms $\Sigma_{x}\left(G\right)$ of the genotype network? From a sufficiency direction, we show that there exist a few graphical operations on the GRNs that produce phenotype-preserving automorphisms. These graphical operations involve permuting/shuffling different sorts of elements in a GRN:(i)The identities of loci Γ, e.g., exchanging edge labels of loci A and B in Figure 4b;(ii)The identities of dummy proteins $\Omega^{\prime }$, e.g., exchanging node labels of proteins 3 and 4 in Figure 4c.

Then, potentially rewiring a given edge (see details in Definitions A1 and A2):(iii)Change the source node of an edge from one stimulus to another stimulus and vice versa, e.g., in Figure 4d, moving an edge pointing from node 1 to node 3 to pointing from node 2. (Note that this operation is not necessarily equivalent to permuting the identities of stimuli since at most only the single focal edge will be affected.)(iv)Move a self-loop at one node to another node and vice versa, for example, re-allocating a self-loop at node 3 to node 4 in Figure 4e.

For the formal proofs, we point the reader to Theorem A1. Additionally, from a necessity direction, one can computationally obtain a partition $\hat{\Theta }$ of the GRNs G that is coarser than the equivalence classes Θ. (A partition P is coarser than another partition $P^{\prime }$ if any group in $P^{\prime }$ is included in some group in P.) Specifically, start with a partition ${\hat{\Theta }}_{0}$ where GRNs with the same resultant phenotype are grouped together. We create a sequence of partitions of G through the following iterative procedure: Given the partition ${\hat{\Theta }}_{i}$, the next partition ${\hat{\Theta }}_{i+1}$ is obtained by further dividing groups into ${\hat{\Theta }}_{i}$ (if needed) such that for each group $\theta \in {\hat{\Theta }}_{i}$ and $\theta^{\prime }\in {\hat{\Theta }}_{i+1}$, any two GRNs in $\theta^{\prime }$ have the same number of neighbors among θ. This iterative procedure is terminated when no further division is required, i.e., ${\hat{\Theta }}_{k+1}={\hat{\Theta }}_{k}$ for some integer *k* (see Figure 6a for an illustrative cartoon of the iterative procedure). We then have $\hat{\Theta }={\hat{\Theta }}_{k}$ to be our desired partition of GRNs.

To see why the proposed iterative procedure generates a coarser partition $\hat{\Theta }$ than the equivalence classes Θ of GRNs, we stress that the equivalence classes under automorphisms always form an equitable partition. A partition $P={\left\{P_{i}\right\}}_{i=1}^{m}$ of nodes of a graph is equitable [81] if for every $P_{i},P_{j}\in P$, any two nodes u,v in group $P_{i}$ have the same number of neighbors in $P_{j}$ (Figure 6b). Since GRNs in an equivalence class θ∈Θ must have the same amount of neighbors for each different phenotype, we inductively show that any two GRNs $g_{1},g_{2}\in \theta$ are never separated during the iterative procedure that generates $\hat{\Theta }$ (see Theorem A2). Therefore, any equivalence class θ∈Θ must be included in a computationally acquired group $\hat{\theta }\in \hat{\Theta }$.

Figure 7 demonstrates the coarser partition $\hat{\Theta }$ generated by the iterative procedure for an arbitrary toy example. The obtained $\hat{\Theta }$ contains 154 groups of GRNs, and the size of groups ranges from 2 to 96. We also count the number of different kinds of GRNs that can not be transformed through graphical operations (i) and (ii), and this number varies from 1 to 4 in our example $\hat{\Theta }$. Moreover, for every group in $\hat{\Theta }$, we observe that those different kinds of GRNs can be related by changing the stimulus that an edge is pointing from and re-allocating self-loops (e.g., see Figure 7b). $\hat{\Theta }$ is thus not simply a coarser partition than the equivalence classes; according to (i)–(iv), we know that groups in $\hat{\Theta }$ are exactly the equivalence classes Θ. This arguably general toy example implicates that there is no need for other graphical operations to determine the equivalence classes of GRNs.

As a result, we conjecture that all the phenotype-preserving automorphisms $\Sigma_{x}\left(G\right)$ of the genotype network can be generated by combining graphical operations (i) to (iv) on the GRNs. In other words, two GRNs $g_{1}$ and $g_{2}$ belong to the same equivalence class if and only if, after removing all the self-loops and merging stimuli $\Omega_{0}$ into a single node, there exist permutations of loci Γ and dummy proteins $\Omega^{\prime }$ that jointly transform $g_{1}$ into $g_{2}$. This condition reconciles with the concept of isomorphisms between graphs. Whereas an automorphism is a mapping of nodes such that a graph preserves itself, an isomorphism is a mapping of nodes that transform one graph into another. We will borrow the terminology and call the two permutations of Γ and $\Omega^{\prime }$ together a *phenotype-preserving isomorphism* from $g_{1}$ to $g_{2}$.

### 3.3. Algorithmic Construction of the Mesoscopic Backbone of GRN Fitness Landscape

Next, we investigate algorithmic approaches to construct the mesoscopic backbone of a fitness landscape based on equivalence classes, where a representative GRN replaces all other GRNs in an equivalence class due to their identical role. In particular, the desired algorithm must (a) acquire the equivalence classes Θ from scratch and (b), for a representative GRN in any equivalence class, count the number of its mutational neighbors in other equivalence classes and also within the class it belongs to.

To avoid any confusion, we emphasize that, although drawing mutational connections between equivalence classes Θ can be achieved by grouping mega-nodes in the genotype network *G*, this naive exercise is unsuitable. First and foremost, grouping mega-nodes demands prior knowledge of the genotype network itself, but its construction is computationally heavy. Second, in contrast to coarse-graining nodes in a graph where the groups of nodes are pre-specified, listing all GRNs in an equivalence class requires examining pairs of GRNs and assuring a phenotype-preserving isomorphism between them after removing self-loops and merging stimuli. Determining the equivalence classes Θ from all the GRNs G=V(G) can thus be costly as well. These reasons again show the value of the equivalence classes Θ, which consolidate GRNs into their equivalent representatives.

Here, we present a bottom-up approach that enumerates each equivalence class of GRNs and simultaneously computes the number of mutational connections among them. To begin, recall from Section 2.2 that a mutation from a GRN $g_{1}\in G$ to another $g_{2}\in G$ corresponds to rewiring a single edge in $g_{1}$, where $g_{1}$ may rewire a self-loop/non-self-loop edge to a self-loop/non-self-loop edge in $g_{2}$. We observe that the number of non-self-loop edges in mutational neighbors $g_{1}$ and $g_{2}$ differ at most by one. We denote by $\Gamma^{\prime }\left(g\right)$ the loci representing the non-self-loop edges in the GRN *g*, and $|\Gamma^{\prime }\left(g\right)|$ the number of those non-self-loop edges. In other words, given equivalence classes $\theta ,\theta^{\prime }\in \Theta$ and representative GRNs g∈θ and $g^{\prime }\in \theta^{\prime }$, *g* has no mutational neighbors in $\theta^{\prime }$ if $\left|\right|\Gamma^{\prime }\left(g\right)|-|\Gamma^{\prime }\left(g^{\prime }\right)\left|\right|>1$.

We can therefore build the mesoscopic backbone by incrementally examining each equivalence class with an increasing number of non-self-loop edges in the representative GRN. This strategy is envisioned in Figure 8, where the backbone can be viewed as “layers” of equivalence classes of GRNs. Let $\Theta_{k}$ be the set of equivalence classes where for every $\theta \in \Theta_{k}$, the representative GRN g∈θ has exactly *k* non-self-loop edges, $|\Gamma^{\prime }\left(g\right)|=k$. We start with layer $\Theta_{0}$, which consists of the only equivalence class with no non-self-loop edges. Then, with layers $\Theta_{0},\Theta_{1},\ldots ,\Theta_{k}$ and all the mutational connections among them, we will find the equivalence classes in the next layer $\Theta_{k+1}$ and their mutational connections with layer $\Theta_{k}$ and within themselves up until k=|Γ|, where all the edges are non-self-loops.

To be more precise, we introduce the concept of $M^{+}$ neighborhood: For any GRN g∈G, denote by $M^{+}\left(g\right)$ the mutational neighbors of *g* that have one more non-self-loop edge than *g*. $M^{+}$ neighborhoods are sufficient to capture the relationship between two mutational neighbors *g* and $g^{\prime }$:If $g^{\prime }$ has one more non-self-loop edge than *g*, then $g^{\prime }\in M^{+}\left(g\right)$;If $g^{\prime }$ has one less non-self-loop edge than *g*, then we have $g\in M^{+}\left(g^{\prime }\right)$;If $g^{\prime }$ has the same number of non-self-loop edges as *g*, and then they share a common mutational neighbor $g^{\prime \prime }$, where the only different edge between *g* and $g^{\prime }$ is rewired to a self-loop and thus $g,g^{\prime }\in M^{+}\left(g^{\prime \prime }\right)$.

The mutational connections between equivalence classes can hence be uncovered by examining the $M^{+}$ neighborhood of the representative GRNs. Moreover, the $M^{+}$ neighborhood of representative GRNs in layer $\Theta_{k}$ reveals the equivalence classes in layer $\Theta_{k+1}$ because any GRN must have a mutational neighbor with one less non-self-loop edge. All that remains is to join different $M^{+}$ neighbors into equivalence classes. In particular:(A)For an equivalence class $\theta \in \Theta_{k}$ and its representative GRN g∈θ, under what condition will $g_{1}^{\prime },g_{2}^{\prime }\in M^{+}\left(g\right)$ belong to the same equivalence class in layer $\Theta_{k+1}$?(B)For two distinct equivalence classes $\theta_{1},\theta_{2}\in \Theta_{k}$ and their representative GRNs $g_{1}\in \theta_{1}$ and $g_{2}\in \theta_{2}$, under what condition will $g_{1}^{\prime }\in M^{+}\left(g_{1}\right)$ and $g_{2}^{\prime }\in M^{+}\left(g_{2}\right)$ belong to the same equivalence class in layer $\Theta_{k+1}$?

For our ease of illustration, we hereafter choose the GRNs *g*, $g_{1}$, $g_{2}$, $g_{1}^{\prime }$ and $g_{2}^{\prime }$ such that only one stimulus node is incident to out-going edges.

To address (A), let $g_{1}^{\prime },g_{2}^{\prime }\in M^{+}\left(g\right)$ belong to the same equivalence class, so there is a phenotype-preserving isomorphism π from $g_{1}^{\prime }$ to $g_{2}^{\prime }$ after self-loop removal. Recalling from Section 2.2, $e_{g}\left(\gamma \right)=\left(u,v\right)$ denotes that “the source–target pair of edge γ is (u,v) in GRN *g*.” Furthermore, we write $e_{g_{1}^{\prime }}\left(\gamma_{1}\right)=\left(u_{1},v_{1}\right)$ and $e_{g_{2}^{\prime }}\left(\gamma_{2}\right)=\left(u_{2},v_{2}\right)$, where $\gamma_{1}$ and $\gamma_{2}$ are the non-self-loop edges “added” to *g* that forms $g_{1}^{\prime }$ and $g_{2}^{\prime }$, respectively. A few observations follow:There is an integer *p* such that $\pi^{p}\left(\gamma_{1}\right)=\gamma_{1}$ and $\left(\pi^{p}\left(u_{1}\right),\pi^{p}\left(v_{1}\right)\right)=\left(u_{1},v_{1}\right)$;There is another integer q<p such that $\pi^{q}\left(\gamma_{1}\right)=\gamma_{2}$ and $\left(\pi^{q}\left(u_{1}\right),\pi^{q}\left(v_{1}\right)\right)=\left(u_{2},v_{2}\right)$;$e_{g_{2}^{\prime }}\left(\pi^{k}\left(\gamma_{1}\right)\right)=\left(\pi^{k}\left(u_{1}\right),\pi^{k}\left(v_{1}\right)\right)$ for k=1,2,…,q;$e_{g_{2}^{\prime }}\left(\pi^{k}\left(\gamma_{1}\right)\right)\neq \left(\pi^{k}\left(u_{1}\right),\pi^{k}\left(v_{1}\right)\right)$ for k=q+1,q+2,…,p;For any locus γ and non-self-loop source–target pair (u,v) such that $\left(\gamma ,u,v\right)\neq (\pi^{k}\left(\gamma \right),\pi^{k}\left(u_{1}\right),\pi^{k}\left(v_{1}\right))$ for 0≤k≤q−1, we have $e_{g}\left(\pi \left(\gamma \right)\right)=\left(\pi \left(u\right),\pi \left(v\right)\right)$ if and only if $e_{g}\left(\gamma \right)=\left(u,v\right)$.

We detail the reasoning behind these observations in Lemma A1–A3. Critically, our fifth observation implies that, after self-loop removal, the isomorphism π between $g_{1}^{\prime }$ and $g_{2}^{\prime }$ is in fact a phenotype-preserving automorphism of a subgraph $\bar{g}$ of the GRN *g*. In addition, observations 3. and 4. show that those edges in *g*—but not in $\bar{g}$—are sequentially mapped from one to another via this automorphism π, i.e., $\Gamma^{\prime }\left(g\right)⧵\Gamma^{\prime }\left(\bar{g}\right)={\left\{\pi^{k}\left(\gamma_{1}\right)\right\}}_{k=1}^{q-1}$, and they bridge the newly added edges $\gamma_{1}$ and $\gamma_{2}=\pi^{q}\left(\gamma_{1}\right)$. We show that the converse is also true (see Theorem A3): After self-loop removal, if we find a phenotype-preserving automorphism π of a subgraph $\bar{g}$ of *g* where $\gamma_{1}$ is consecutively mapped to $\gamma_{2}$ through the edge differences $\Gamma^{\prime }\left(g\right)⧵\Gamma^{\prime }\left(\bar{g}\right)$, π is guaranteed a phenotype-preserving isomorphism from $g_{1}^{\prime }$ to $g_{2}^{\prime }$.

The sufficient and necessary condition for two $M^{+}$ neighbors of *g* to be in the same equivalence class, intriguingly, lies in the phenotype-preserving automorphisms of subgraphs of the representative GRN *g*. Here, we demonstrate a few simple examples in Figure 9a. In the top row, an automorphism of *g* directly maps between the two additional edges $\left(u_{1},v_{1}\right)=\left(3,5\right)$ and $\left(u_{2},v_{2}\right)=\left(1,4\right)$. In the middle row, the two edges $\left(u_{1},v_{1}\right)=\left(1,2\right)$ are consecutively mapped to $\left(u_{2},v_{2}\right)=\left(3,4\right)$ through edge (2,3), and $\left(u_{2},v_{2}\right)$ is consecutively mapped back to $\left(u_{1},v_{1}\right)$ through the non-edge (4,1), so we have q=2 and p=4. As a mixture of both, in the bottom row, $\left(u_{1},v_{1}\right)=\left(2,5\right)$ is consecutively mapped to $\left(u_{2},v_{2}\right)=\left(3,6\right)$ through edge (1,4), and this isomorphism is exactly an automorphism of a subgraph $\bar{g}$ of *g* where edge (1,4) is removed.

Switching gears to the remaining question (B), suppose that $g_{1}$ and $g_{2}$ are the representative GRN in two different equivalence classes where $|\Gamma^{\prime }\left(g_{1}\right)\left|=\right|\Gamma^{\prime }\left(g_{2}\right)|$ and that $g_{1}^{\prime }\in M^{+}\left(g_{1}\right)$ and $g_{2}^{\prime }\in M^{+}\left(g_{2}\right)$ belong to the same equivalence class. Let $\gamma_{1}$ and $\gamma_{2}$ be the newly added edges to $g_{1}$ and $g_{2}$ that generate $g_{1}^{\prime }$ and $g_{2}^{\prime }$, respectively, where $e_{g_{1}^{\prime }}\left(\gamma_{1}\right)=\left(u_{1},v_{1}\right)$ and $e_{g_{2}^{\prime }}\left(\gamma_{2}\right)=\left(u_{2},v_{2}\right)$, and let π be a phenotype-preserving isomorphism from $g_{1}^{\prime }$ to $g_{2}^{\prime }$ after self-loop removal. We observe that applying the permutation π on $g_{1}$ transforms it into another GRN ${\tilde{g}}_{1}$ in the same equivalence class. Since $g_{1}$ simply has one less edge $\gamma_{1}$ than $g_{1}^{\prime }$, and ${\tilde{g}}_{1}$ and $g_{2}^{\prime }$ only differ by a missing edge $\pi (\gamma_{1})$. Namely, we have $g_{2}^{\prime }\in M^{+}\left({\tilde{g}}_{1}\right)$ with the additional edge $e_{g_{2}^{\prime }}\left(\pi \left(\gamma_{1}\right)\right)=\left(\pi \left(u_{1}\right),\pi \left(v_{1}\right)\right)$. Moreover, since $g_{2}^{\prime }$ also belongs to the $M^{+}$ neighborhood of $g_{2}$ with the additional edge $e_{g_{2}^{\prime }}\left(\gamma_{2}\right)=\left(u_{2},v_{2}\right)$, by removing both the extra edges from $g_{2}^{\prime }$, we find a GRN $g^{\prime \prime }$ such that ${\tilde{g}}_{1},g_{2}\in M^{+}\left(g^{\prime \prime }\right)$.

We again present an illustrative example in Figure 9b. Here, a GRN ${\tilde{g}}_{1}$ in the equivalence class of $g_{1}$ can be found via the isomorphism π between $g_{1}^{\prime }$ and $g_{2}^{\prime }$. We note that the newly added edge $\left(u_{1},v_{1}\right)=\left(4,1\right)$ is transformed into $\left(\pi \left(u_{1}\right),\pi \left(v_{1}\right)\right)=\left(3,4\right)$ in $g_{2}^{\prime }$, which is missing in ${\tilde{g}}_{1}$. Removing both $\left(\pi \left(u_{1}\right),\pi \left(v_{1}\right)\right)=\left(3,4\right)$ and $\left(u_{2},v_{2}\right)=\left(3,1\right)$ from $g_{2}^{\prime }$ produces a GRN $g^{\prime \prime }$, which is a common neighbor of $g_{2}$ and ${\tilde{g}}_{1}$ with one less non-self-loop edge.

Our observation resolves the necessary condition of (B): For the representative GRNs of two different equivalence classes $g_{1}\in \theta_{1}$ and $g_{2}\in \theta_{2}$, if their $M^{+}$ neighbor $g_{1}^{\prime }\in M^{+}\left(g_{1}\right)$ and $g_{2}^{\prime }\in M^{+}\left(g_{2}\right)$ belong to the same equivalence class, then we can always find two GRNs ${\tilde{g}}_{1}$ and $g^{\prime \prime }$ such that (a) ${\tilde{g}}_{1}$ falls into the equivalence class of $g_{1}$, and (b) ${\tilde{g}}_{1}$ and $g_{2}$ are $M^{+}$ neighbors of $g^{\prime \prime }$. Moreover, the converse is true as well (Theorem A4). Therefore, whether the $M^{+}$ neighborhood of $g_{1}$ and $g_{2}$ reveal a common equivalence class depends on the existence of a GRN $g^{\prime \prime }$ that both the equivalence classes $\theta_{1}$ and $\theta_{2}$ are rooted from.

Our strategy to build the mesoscopic backbone is now complete, and here, we detail our algorithm that incrementally generates the equivalence classes Θ of GRNs and establishes the mutational connections among them. Suppose that we have already built layers of equivalence classes $\Theta_{0},\Theta_{1},\ldots ,\Theta_{k}$ and determined the mutational connections among them. For each representative GRN *g* in layer $\Theta_{k}$ and every $g^{\prime }\in M^{+}\left(g\right)$, we will view $g^{\prime }$ as the combination of *g* and an additional, non-self-loop edge $e_{g^{\prime }}\left(\gamma \right)=\left(u,v\right)$, for which we write $g^{\prime }=g⊕\left(\gamma ,u,v\right)$. All such combinations form a collection of $M^{+}$ neighbors of the representative GRNs in layer $\Theta_{k}$, for which we abuse the notation $M^{+}\left(\Theta_{k}\right)$.

We initially put each $g^{\prime }\in M^{+}\left(\Theta_{k}\right)$ into an individual group, and we define a collection of operations Φ that join groups of $M^{+}$ neighbors:(I)For every representative GRN *g* in $\Theta_{k}$ and every phenotype-preserving automorphism σ of *g*, there is an operation $\psi_{g,\sigma }$ that joins together the groups of $g_{1}^{\prime }=g⊕\left(\gamma ,u_{1},v_{1}\right)$ and $g_{2}^{\prime }=g⊕\left(\gamma ,u_{2},v_{2}\right)$, where $u_{1},u_{2}\in \Omega_{0}$ and $v_{2}=\sigma \left(v_{1}\right)$;(II)For every representative GRN *g* in $\Theta_{k}$ and every phenotype-preserving automorphism $\bar{\sigma }$ of each subgraph $\bar{g}$ of *g* such that the edge differences $\Gamma^{\prime }\left(g\right)⧵\Gamma^{\prime }\left(\bar{g}\right)$ are sequentially connected via $\bar{\sigma }$, there is an operation $\phi_{g,\bar{g},\bar{\sigma }}$ that joins together the groups of $g_{1}^{\prime }=g⊕\left(\gamma_{1},u_{1},v_{1}\right)$ and $g_{2}^{\prime }=g⊕\left(\gamma_{2},u_{2},v_{2}\right)$, where automorphism $\bar{\sigma }$ consecutively transforms edge $\gamma_{1}$ into $\gamma_{2}$ through $\Gamma^{\prime }\left(g\right)⧵\Gamma^{\prime }\left(\bar{g}\right)$;(III)For every representative GRN $g^{\prime \prime }$ in $\Theta_{k-1}$ and each ${\tilde{g}}_{1}=g^{\prime \prime }⊕\left(\gamma_{1}^{\prime \prime },u_{1}^{\prime \prime },v_{1}^{\prime \prime }\right)$ and ${\tilde{g}}_{2}=g^{\prime \prime }⊕\left(\gamma_{2}^{\prime \prime },u_{2}^{\prime \prime },v_{2}^{\prime \prime }\right)$ in two different equivalence classes $\theta_{1}$ and $\theta_{2}$, such that we have phenotype-preserving isomorphisms $\pi_{1}$/$\pi_{2}$ from ${\tilde{g}}_{1}$/${\tilde{g}}_{2}$ to the representative GRN $g_{1}$/$g_{2}$ after self-loop removal, there is an operation $\varphi_{g,{\tilde{g}}_{1},{\tilde{g}}_{2}}$ that joins together the groups of $g_{1}^{\prime }=g_{1}⊕\left(\pi_{2}\left(\gamma_{2}^{\prime \prime }\right),\pi_{2}\left(u_{2}^{\prime \prime }\right),\pi_{2}\left(v_{2}^{\prime \prime }\right)\right)$, and $g_{2}^{\prime }=g_{2}⊕\left(\pi_{1}\left(\gamma_{1}^{\prime \prime }\right),\pi_{1}\left(u_{1}^{\prime \prime }\right),\pi_{1}\left(v_{1}^{\prime \prime }\right)\right)$.

The resulting groups of $M^{+}$ neighbors, after applying the joining operations Φ, constitute the equivalence classes in the next layer $\Theta_{k+1}$. We hereafter denote by $M_{\Phi }^{+}\left(\theta^{\prime }\right)$ the corresponding consequent group of an equivalence class $\theta^{\prime }\in \Theta_{k+1}$. We then choose an arbitrary $M^{+}$ neighbor in $M_{\Phi }^{+}\left(\theta^{\prime }\right)$ as the representative GRN of the equivalence class $\theta^{\prime }$, such that only one stimulus node is incident to out-going edges in the chosen representative GRN.

The joining operations Φ further provide useful information to count the number of mutation neighbors that a representative GRN g∈θ in layer $\Theta_{k}$ has among any equivalence class $\theta^{\prime }$, which we will denote by $A_{g}\left(\theta^{\prime }\right)$. Let us first consider $\theta^{\prime }\in \Theta_{k+1}$. For any ${\tilde{g}}^{\prime }\in M_{\Phi }^{+}\left(\theta^{\prime }\right)$, ${\tilde{g}}^{\prime }$ is a mutational neighbor of *g* if it can be viewed as a combination of *g* and an arbitrary extra non-self-loop edge, and hence
(1)$$A_{g}\left(\theta^{\prime }\right)=\left|M_{\Phi }^{+}\left(\theta^{\prime }\right)\cap M^{+}\left(g\right)\right|\;,\;\mathrm{for}\;\theta^{\prime }\in \Theta_{k+1}\;.$$
Note that, in this case, $A_{g}\left(\theta^{\prime }\right)$ is easily acquired when building up the layer $\Theta_{k+1}$ through Φ.

Second, for $\theta^{\prime }\in \Theta_{k-1}$, $A_{g}\left(\theta^{\prime }\right)$ can be computed given $A_{g^{\prime }}\left(\theta \right)$, where $g^{\prime }$ is the representative GRN of $\theta^{\prime }$. Since the equivalence classes Θ generate an equitable partition of the genotype network *G* (see Section 3.2), we have $A_{g}\left(\theta^{\prime }\right)\times \left|\theta \right|=A_{g^{\prime }}\left(\theta \right)\times \left|\theta^{\prime }\right|$ equal to the total number of mutational connections between θ and $\theta^{\prime }$. Moreover, the size of the equivalence class θ is (see Appendix D)
(2)$$\left|\theta \right|=\frac{|\Pi^{\prime }|}{|\Sigma^{\prime }\left(g\right)|}\times n_{l}\left(|\Gamma |-k\right)\times m_{s}\left(g\right)\times r\left(g\right)\;,$$
where (a) we denote by $\Pi^{\prime }$ the set of all permutations of dummy proteins $\Omega^{\prime }$ and denote by $\Sigma^{\prime }\left(g\right)$ the set of automorphisms of the representative GRN *g* after self-loop removal that only permutes $\Omega^{\prime }$; (b) $n_{l}\left(|\Gamma |-k\right)$ is the number of ways to allocate |Γ|−k labeled self-loops among the proteins Ω; (c) $m_{s}\left(g\right)$ is the number of ways to re-distribute the edges pointing from stimuli $\Omega_{0}$ in *g*; and (d) r(g) is the number of ways to divide loci Γ into self-loops, non-self-loop edges pointing from stimuli, and others. As a result,
(3)$$A_{g}\left(\theta^{\prime }\right)=A_{g^{\prime }}\left(\theta \right)\times \frac{|\Sigma^{\prime }\left(g\right)|}{|\Sigma^{\prime }\left(g^{\prime }\right)|}\times \frac{n_{l}\left(|\Gamma |-k+1\right)m_{s}\left(g^{\prime }\right)r\left(g^{\prime }\right)}{n_{l}\left(|\Gamma |-k\right)m_{s}\left(g\right)r\left(g\right)}\;,\;\mathrm{for}\;\theta^{\prime }\in \Theta_{k-1}\;.$$

Third, we turn to the case where $\theta^{\prime }\in \Theta_{k}$ but $\theta^{\prime }\neq \theta$. Recall that, if any ${\tilde{g}}^{\prime }\in \theta^{\prime }$ is a mutational neighbor of *g*, then there is a GRN ${\tilde{g}}^{\prime \prime }$ in layer $\Theta_{k-1}$, where $g,{\tilde{g}}^{\prime }\in M^{+}\left({\tilde{g}}^{\prime \prime }\right)$, and such ${\tilde{g}}^{\prime \prime }$ is unique up to arbitrary self-loop re-allocation. Additionally, the extra edge in *g* and ${\tilde{g}}^{\prime }$ must correspond to the same locus, so
(4)$$A_{g}\left(\theta^{\prime }\right)=\sum_{\theta^{\prime \prime }\in \Theta_{k-1}}\frac{A_{g}\left(\theta^{\prime \prime }\right)}{n_{l}\left(1\right)}\times A_{g^{\prime \prime }}\left(\theta^{\prime }\right)\;,\;\mathrm{for}\;\theta^{\prime }\in \Theta_{k-1},\;\theta^{\prime }\neq \theta \;,$$
in which we use $g^{\prime \prime }$ to be the representative GRN of equivalence class $\theta^{\prime \prime }$. Lastly, if $\theta^{\prime }=\theta$, we also need to include the scenario that the mutational neighbor ${\tilde{g}}^{\prime }$ of *g* is generated by rewiring a self-loop to another self-loop. Therefore,
(5)$$A_{g}\left(\theta \right)=\left(|\Gamma |-k\right)\times \left(n_{l}\left(1\right)-1\right)+\sum_{\theta^{\prime \prime }\in \Theta_{k-1}}\frac{A_{g}\left(\theta^{\prime \prime }\right)}{n_{l}\left(1\right)}\times \left[A_{g^{\prime \prime }}\left(\theta \right)-1\right]\;.$$

In Algorithm 1, we summarize our proposed approach that constructs the mesoscopic backbone. It is apparent that the core of our algorithm is determining the joining operations Φ for a given layer $\Theta_{k}$. This task can be achieved by pre-computing the phenotype-preserving automorphisms of every representative GRN once it is chosen. In addition, since these joining operations reflect the mutational neighbors and the phenotype-preserving isomorphisms in previous layers, the type-(III) Φ for layer $\Theta_{k}$ is generated as a composition of the already uncovered operations. Furthermore, the remaining Φ of type (II) consists of combinations of the uncovered joining operations and the newly computed automorphisms of representative GRNs in layer $\Theta_{k}$. As a result, the only prerequisite in our proposed algorithm is producing the phenotype-preserving automorphisms of a GRN.
**Algorithm 1** Constructing the underlying space of a fitness landscape of GRNs**Require:** The fixed underlying collections of loci Γ and proteins Ω of GRNs
**Ensure:** The representative GRN $g_{\theta }$ of each equivalence class θ∈Θ, and its number of mutational neighbors $A_{g_{\theta }}\left(\theta^{\prime }\right)$ in any equivalence class $\theta^{\prime }\in \Theta$1:k←0                            ▹ initialization2:$g_{\theta_{0}}\leftarrow$ a GRN with no self-loop, where $\theta_{0}$ is the only equivalence class in layer $\Theta_{0}$3:Store the phenotype-preserving automorphisms $\Sigma_{x}\left(g_{\theta_{0}}\right)$.4:Compute $A_{g_{\theta_{0}}}\left(\theta_{0}\right)$ via Equation (Equation 5).5:**while**k<|Γ|**do**                    ▹ incrementally find Θ6:    Construct and store the joining operations Φ for layer $\Theta_{k}$.7:    $M_{\Phi }^{+}\leftarrow$ grouping of $M^{+}\left(\Theta_{k}\right)$ acted by Φ8:    $\Theta_{k+1}$ corresponds to the groups in $M_{\Phi }^{+}$.9:    **for all** $\theta^{\prime }\in \Theta_{k+1}$ **do**10:        $g_{\theta^{\prime }}\leftarrow$ a GRN in $M_{\Phi }^{+}\left(\theta^{\prime }\right)$          ▹ choose the representative GRN11:        Store the phenotype-preserving automorphisms $\Sigma_{x}\left(g_{\theta^{\prime }}\right)$.12:    **end for**13:    **for all** $\theta \in \Theta_{k},\theta^{\prime }\in \Theta_{k+1}$**do**     ▹ count the number of mutational neighbors14:        Compute $A_{g_{\theta }}\left(\theta^{\prime }\right)$ and $A_{g_{\theta^{\prime }}}\left(\theta \right)$ via Equations (Equation 1) and (Equation 3).15:    **end for**16:    **for all** $\theta_{1}^{\prime },\theta_{2}^{\prime }\in \Theta_{k+1}$ **do**17:        Compute $A_{g_{\theta_{1}^{\prime }}}\left(\theta_{2}^{\prime }\right)$ via Equations (Equation 4) and (Equation 5).18:    **end for**19:    k←k+120:**end while**21:Set any remaining, not computed $A_{g_{\theta }}\left(\theta^{\prime }\right)$ to zero.


## 4. Conclusions

In this work, we integrate mechanistic knowledge of how phenotypes are computed from genotypes via regulatory interactions into fitness landscape models. The resulting family of fitness landscape models features flexibility for tunable ruggedness and accessibility among phenotypes. Furthermore, we introduce the concept of equivalence classes of GRNs, where GRNs of the same phenotype and with similar structural positions in the genotype network are coarse-grained into a group. These equivalence classes of GRNs lead to a compact and informative description of the fundamental space of a fitness landscape. Using this coarse-graining, we develop a bottom-up, efficient algorithm for constructing the underlying space of a fitness landscape based on the equivalence classes. Critically, this algorithm does not require pre-computing the genotype network and therefore permits the exploration of substantially larger GRNs.

Naively, ruggedness and accessibility would seem to be contradictory characteristics of a fitness landscape. Indeed, reciprocal sign epistasis has been shown to yield a strong influence on a landscape’s ruggedness and was regarded as an impediment to evolutionary accessibility when first introduced [2,32,55]. Nevertheless, recent studies suggest that fitness landscape models most closely aligned with empirical observations show that sign epistasis (and thus ruggedness) can co-exist with accessibility [63,82]. In addition to demonstrating that ruggedness and accessibility are not mutually exclusive, our model is compatible with three additional empirical observations. First, GRNs result in high dimensional genotype–phenotype maps [63]. Second, selection acts on the superposition of mutations and the background GRN rather than a few pairs of mutations [60]. Third, and perhaps most importantly, a GRN may experience a series of neutral mutations and then evolve into a nearby phenotype [3,8,83,84]. The accessibility induced in fitness landscapes of GRNs via neutral evolution agrees with the phenomenon of punctuated equilibrium/epochal evolution [23,85,86].

Our derived equivalence classes for GRNs provide a novel, mesoscopic, and optimally descriptive skeleton of a fitness landscape. Neither the genotypic space nor the phenotypic space alone fully characterize a fitness landscape; however, models with even a relatively simple genotype–phenotype map are computationally intensive because they must retain all plausible genotypes [70,73,74,75]. Intuitively, the complexity of a genotype–phenotype map can be reduced by combining similar phenotypes into high-level descriptors [87]. The equivalence classes of GRNs, on the other hand, serve as an intermediate level between the genotypic and phenotypic space, which provides an optimal coarse-graining that encodes all necessary information to predict the evolutionary trajectory on the fitness landscape.

We argue that our proposed algorithm for coarse-graining GRN fitness landscapes is more efficient than brute-force approaches. First, because we consolidate an equivalence class into a single representative GRN, our method is less costly in memory and requires fewer computations when finding mutational neighbors. Second, suppose all plausible GRNs were organized into layers by the number of non-self-loop edges (see Section 3.3), every layer would still super-exponentially contain many GRNs. Our algorithm instead finds the equivalence classes in each layer iteratively. To construct the (k+1)-th layer, we only have to exhaust the representative GRNs in the *k*-th layer–along with any plausible additional non-self-loop edge(s)–this amount will be significantly fewer than the number of GRNs in the (k+1)-th layer. Lastly, existing heuristics for graph automorphisms [88,89] can be used to produce the phenotype-preserving automorphisms of the representative GRNs, which is the only prerequisite when joining together different GRN–edge pairs. Because the set of automorphisms becomes more limited as the complexity of GRNs increases, we expect only a minor overhead in the joining procedure as compared to the exhaustive, brute-force approach.

Despite our model being constrained to the pathway framework of GRNs [77,78] and a few naive assumptions described in Section 2, we believe our methodology to be flexible and, in what follows, we outline some potential directions to extend the framework. First, when GRNs are modeled through more complex computation, e.g., with different logic gates connecting multiple expression activators/suppressors/products, those GRNs that only consist of naive interactions are never excluded. Thus, the current model represents a subset of the complete landscape built by more complex gene regulation. The derived connectivity and accessibility among the naive GRNs still hold, and we expect these topographical features to manifest for complex GRNs if mutations between the simple and complex expressions are permitted. Second, hypergraphs [90] could be used to describe the expression behavior of genes where multiple activators/products appear. Third, stable motif identification [91] and target control [92] for Boolean network models could be used to explore the phenotypes of mutational neighbors of a focal complex GRN. Lastly, our methodologies are likely applicable to other classes of genotype–phenotype maps [93,94]. In particular, once the mapping and the genotype network are determined, one can simply follow the proposed iterative procedure (Figure 6) to obtain a genotype partition coarser than the equivalence classes.

More broadly, this work showcases the potential of combining biological computation across different scales along the hierarchy of living systems. Computing biological functionality on the organism level with genotype–phenotype mapping provides a blueprint of the overall fitness landscape, where evolutionary processes occur/compute on the population level. Furthermore, several intriguing perspectives arise from the proposed mesoscopic backbone if we consider evolution to be a random walk on the fitness landscape. The process of evolution not only manifests genotypes with higher fitness values but also reveals genotypes whose mutational neighbors are more fit [19,23,78]; in other words, the prevalence of different genotypes would reflect the connection counts between equivalence classes of GRNs. In addition, these “connection counts” could become associated with an analogous theory of computation in evolution that addresses questions such as how likely a genotype in an equivalence class is to evolve into a specified phenotype, as well as how likely it is to “reset” to another genotype in the same equivalence class and recover its position in the fitness landscape. 

## Figures and Tables

**Figure 1 entropy-24-00622-f001:**
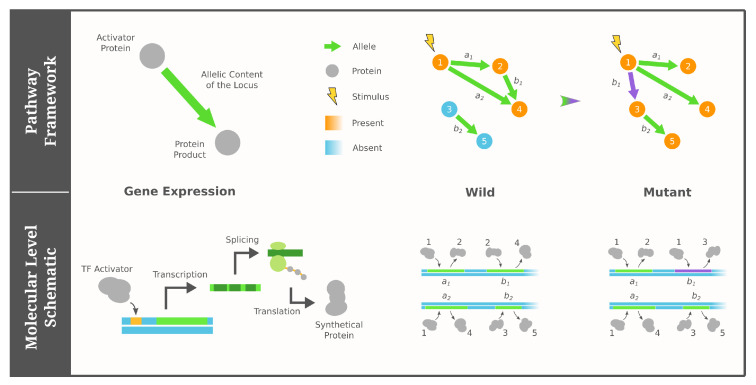
Cartoon illustration of the pathway framework of GRNs adapted from [77,78]. Under our four simplified assumptions, a GRN (genotype) consists of a fixed number of proteins as nodes and a constant number of directed edges depicting the activator/product pairs of genes. The phenotype is modeled as the Boolean states of proteins (colored), which are determined by their reachability from the external stimulus (lightning icon).

**Figure 2 entropy-24-00622-f002:**
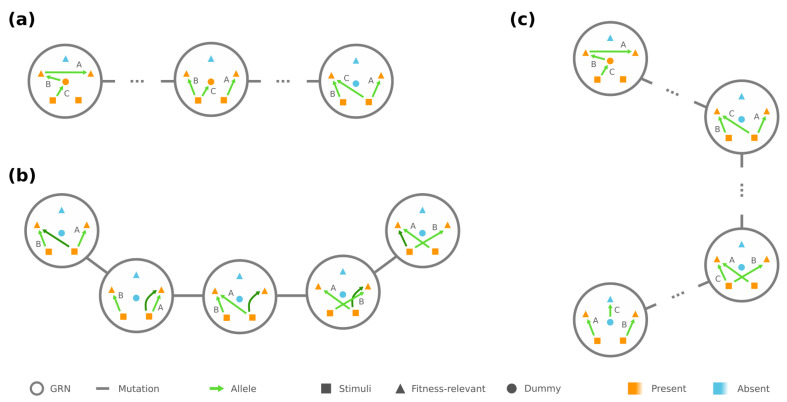
Connectivity exists between all GRNs of the same phenotype. (**a**) Any GRN can be rewired/mutated into a “central” GRN (shown on the right). (**b**) A redundant edge (dark green) makes it feasible to turn any central GRN into another via edge rewiring. (**c**) There is a mutational trajectory between any GRNs of the same phenotype through the central GRNs.

**Figure 3 entropy-24-00622-f003:**
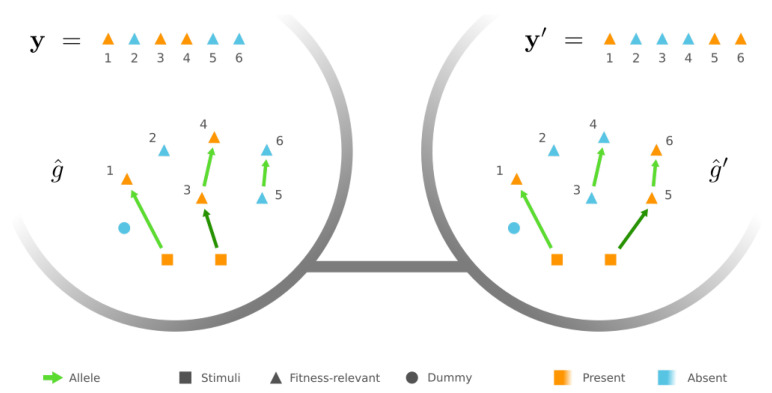
Example of peripheral GRNs connecting two different phenotypes—a peripheral GRN $\hat{g}$ of phenotype y in this example. There is a chain that triggers the presence state of proteins $\Omega_{y}^{+}⧵\Omega_{y^{\prime }}^{+}=\left\{3,4\right\}$. However, the other peripheral GRN ${\hat{g}}^{\prime }$ of phenotype $y^{\prime }$ contains a chain of proteins $\Omega_{y^{\prime }}^{+}⧵\Omega_{y}^{+}=\left\{5,6\right\}$. $\hat{g}$ and ${\hat{g}}^{\prime }$ are mutational neighbors since they only differ by rewiring the dark green edge, i.e., the first edge in either chain.

**Figure 4 entropy-24-00622-f004:**
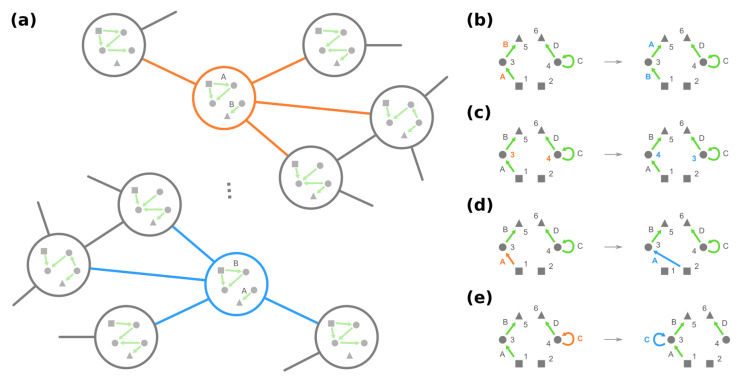
The genotype network has symmetry such that multiple GRNs have similar local neighborhoods, as we demonstrate in (**a**) since the corresponding GRNs only differ by exchanging the role of loci A and B. More formally, these GRNs constitute an equivalence class under phenotype-preserving automorphisms, which can be found by graphical operations of (**b**) permuting loci, (**c**) permuting dummy proteins (circles), (**d**) exchanging edges pointing from two different stimuli (squares), and (**e**) exchanging self-loops at two different nodes.

**Figure 5 entropy-24-00622-f005:**
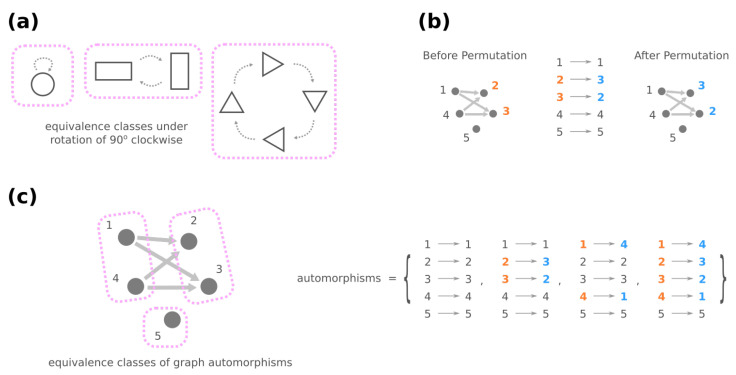
(**a**) As a minimal example, imagine an operation that rotates a geometric object 90 degrees clockwise. The rotation maps one object onto another (dashed arrows), and it leads to equivalence classes where objects are grouped by their symmetry under rotation (pink rectangles). (**b**) An automorphism of a graph is a permutation of nodes that retains the same graph. (**c**) Equivalence classes under graph automorphisms bring together nodes that have similar roles connection-wise in the graph.

**Figure 6 entropy-24-00622-f006:**
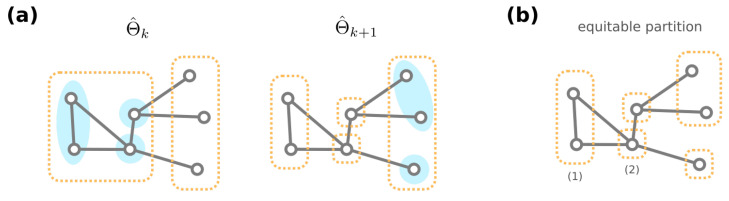
(**a**) Consider a toy example genotype network of GRNs. (Here, we omit the exact content of GRNs.) Given the partition ${\hat{\Theta }}_{k}$, note that mega-nodes in a group (dashed orange rectangle) may share a different number of connections among other groups (blue shaded circles), and they are further divided to generate the next partition ${\hat{\Theta }}_{k+1}$. (**b**) Both the equivalence classes of GRNs and the stationary partition from our iterative procedure are equitable, e.g., each mega-node in group (1) has one connection among (1), another connection with (2), and none with other groups.

**Figure 7 entropy-24-00622-f007:**
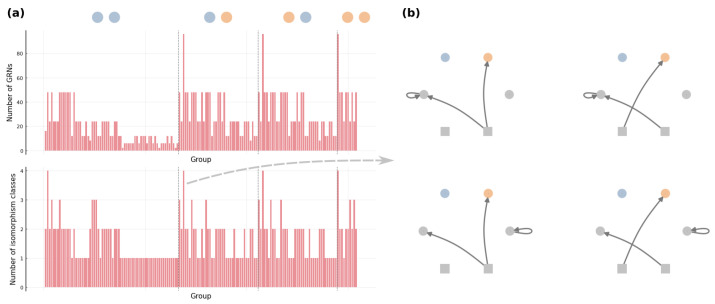
Example partition coarser than the equivalences classes of GRNs. We run the proposed iterative procedure with |Γ|=3 and $|\Omega_{0}\left|=\right|\Omega^{\prime }\left|=\right|\hat{\Omega }|=2$, where stimuli $\Omega_{0}$ are drawn as squares and the present/absent state of fitness-relevant proteins $\hat{\Omega }$ are colored by orange/blue. (**a**) The number of GRNs and the number of isomorphism classes of GRNs in each group of the obtained partition $\hat{\Theta }$, where the dashed lines separate groups of different phenotypes and (**b**) isomorphism classes of GRNs in a group.

**Figure 8 entropy-24-00622-f008:**
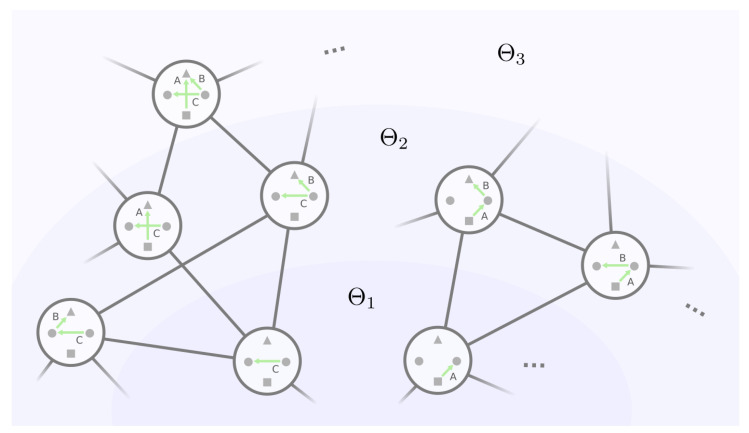
Layering the GRNs by their number of non-self-loop edges. A GRN’s mutational neighbors must fall into the same or the adjacent layers. For ease of illustration, we only show the non-self-loop edges and neglect the protein states in GRNs.

**Figure 9 entropy-24-00622-f009:**
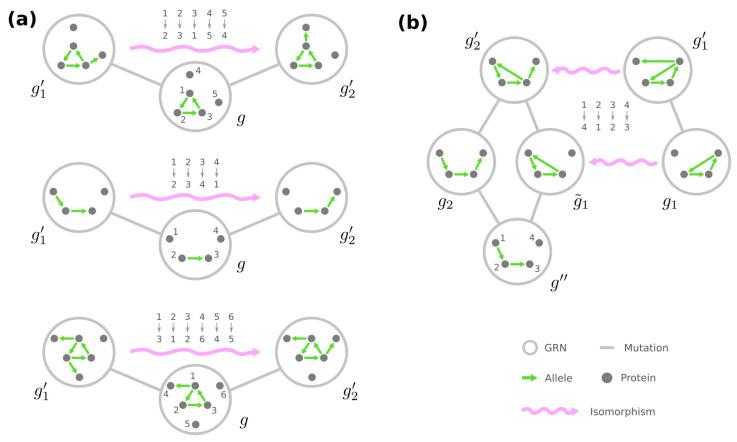
Sufficient conditions that two $M^{+}$ neighbors belong to an equivalence class. For illustration purposes, we only show the dummy proteins and omit the protein states, edge labels and self-loops in (**a**) three examples such that two $M^{+}$ neighbors of a GRN *g* are isomorphic, and (**b**) an example where the $M^{+}$ neighbors $g_{1}^{\prime }$ and $g_{2}^{\prime }$ of GRNs $g_{1}$ and $g_{2}$ in different equivalence classes are isomorphic.

## Data Availability

The authors affirm that all data necessary for confirming the conclusions of the article are present within the article, gures, and tables.

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
