# Peer review of "A Family of Fitness Landscapes Modeled through Gene Regulatory Networks"

_entropy, 2022, doi:10.3390/e24050622_

Round 1

Reviewer 1 Report

Review of "A family of fitness landscapes modeled through gene regulatory networks"

In this manuscript, the authors present a framework for thinking about biological fitness landscapes in terms of a specific model of the genotype-phenotype map based on some assumptions about gene regulation.  They use their framework to show that, under these assumptions, (1) phenotypes are relatively accessible (reachable by mutations that do not decrease fitness) and (2) there is a procedure for efficiently delineating the space of all possible regulatory networks in terms of genotypic and phenotypic connectivity, making use of graph isomorphisms.  The manuscript is detailed, rigorous, and clear (though somewhat dense).  The algorithm for constructing the coarse-grained connectivity (the "mesoscopic backbone") could be very useful for future work on genotype-phenotype maps in this model of gene regulation (and potentially others when the graph automorphisms are known).

The focus of the manuscript remained abstract and mathematical, with limited application of the results to biology, which is reasonable given the venue.  Still, I could have been more convinced that the biological assumptions made are a good starting point for building out the theory.  In particular, the framework assumes very simple regulatory mechanisms, such that the expression of a given gene "can only be activated by one specific protein".  I understand that this simplifying assumption could be necessary to build the framework before adding more realistic details.  But it also seems that more complex combinatorial logic is both common in biology and could fundamentally alter the topology of the spaces that the authors analyze.  A brief discussion about how these assumptions fit into the broader biological picture could be useful for motivating the remainder of the analysis.

Other minor points / typos:

Abstract, "Despite": This felt like an odd word choice.  Isn't it the high dimensionality of landscapes that is the problem?

line 32 "each others"

line 158 "two different phenotype"

line 160 "an stimulus"

line 231, operation iii: Is this equivalent to saying that the identities of the stimuli are permuted?

line 286 "First and for most" -> "First and foremost"

line 400 "groups of after applying"

line 416 "automrphism"

line 433 "typed-(III)" -> "type-(III)" ?

line 455 "may experiences"

line 462 "addition"

line 466 "an common"

line 489, final sentence: This sentence seemed to come out of nowhere.  How does the work address computation across scales?  Just in the sense that the phenotype is a "computation" resulting from the regulatory network?  It feels like this point could be made more clearly.

Reviewer 2 Report

The mapping of genotypes onto phenotypes that determine the biological fitness of an organism is one of the most fundamental endeavors in biology. The structure of this map has major implications for evolution, development, and disease. The mapping from a high-dimensional space of genotypes onto a quantitative phenotype or fitness can be conceptualized with the concept of the fitness or adaptive landscape. Since Sewall Wright introduced fitness landscapes in the early 1930s, they have received considerable attention from theorists interested in understanding how landscape topography affects evolutionary dynamics. In recent years, attention has shifted from theoretical analyses to empirical characterizations of landscapes constructed from experimental data. This shift has been triggered by advances in high-throughput sequencing and chip-based technologies, which have made it possible to assign phenotypes or fitness values to a large number of genotypes. The study of fitness landscapes is therefore a very active area of research in biology. 

In this paper, Yang and Scarpino introduce a new family of fitness landscape based on gene regulatory networks. Gene regulatory circuits occupy a central position in the genotype-phenotype map, because they control when, where, and to what extent genes are expressed, and thus drive fundamental physiological, developmental, and behavioral processes in living organisms as different as bacteria and humans. The authors find that the fitness landscape displays evolutionart-relevant properties such as connectivity and accessibility that are insensitive to the specific fitness function employed determine the topography of the landscape. The authors also find a mesoscopic description of the landscape that while retaining all the properties relevant for evolutionary dynamics it greatly reduces its complexity. I think this paper deserves publication in Entropy. However, I feel that the paper would have a much higher chance of appealing to researchers in the field of empirical fitness landscapes if the writing was made more accessible to researchers with a less strong background in graph theory, if the biological relevance of their models and discussion was better introduced and discussed and if they discussed a bit better how some of the results can be applied to empirical fitness landscapes or even other computational models of genotype-phenotype maps. Besides that I have only a series of minor considerations.

- Line 17: The term genotype network was introduced by Andreas Wagner to replace the term "neutral network" introduced by Schuster and collaborators in 1994. The reason for the new term is that Schuster et al. meant by neutrality the invariance of a well-defined phenotype among all genotypes on a neutral networks. However, neutrality in evolutionary biology implies a change in genotype that is invisible to natural selection because it does not affect fitness. Wagner introduced the term genotype network to avoid that confusion. A more suitable citation could be his book "The Origins of Evolutionary Innovations." 

- Most empirical fitness landscapes are imcomplete. Even the ones cited in the introduction as more comprehensive than earlier studies. They only contain phenotypic or fitness information for a small fraction of all possible genotypes of a given biological system. To my knowledge the only exception is the following paper (Aguilar-Rodriguez et 2017, Nature Ecology and Evolution) that I feel should be cited in the introduction. Also as an example of empirical landscapes with moderate ruggedness in line 33.  

- I also felt that the introduction should mention the models of genotype networks of gene regulatory networks studied in some other papers: Ciliberti et al 2007 PNAS, or Payne and Wagner 2007 BMC Systems Biology for example. It may be interesting to discuss why to choose the pathway framework over the GRN models of these other paper as the basis of a fitness landscape. Pros and cons, etc.

- Section 2.1. I feel like the introduction of the "pathway framework of GRNs" which is the gene regulatory model serving as the basis of their fitness landscape was not enough and I had to consult the previous publications they have on the model. The inclusion of a figure here may also be useful. 

- Section 2.1, second paragraph. I would appreciate a much honest assesment of the biological relevance of the assumptions of their model which almost always violate the properties of real gene regulatory networks. I am especially concerned with assumption c which leads to GRNs that have a very different topology than real ones. 

- Line 106: explain when first introduced what dummy variables are even if it becomes clear later on what they are. 

- Line 189: Figure 4 is introduced before Figure 3.

- Line 451: Only reciprocal sign epistasis is necessary for multi-peaked landscapes but not simple sign epistasios (Poelwijk et al 2011, Journal of Theoretical Biology).

- The mesoscopic structure underlying their fitness landscapes models is analogous to simplifying the complexity of a genotype-phenotype map by focusing on high-level phenotypes that aggregate phenotypes that may have the same or similar fitness values. See for example, Aguilar-Rodriguez et al (2018, Evolution): genotype networks of DNA-binding domains against genotype networks of transcription factor binding sites in that paper. 

- In the discussion I would appreciate a larger discussion of how easy or hard would be to translate their equivalence classes to other computational models of GP maps or even empirical fitness landscapes to get a mesoscopic description of such landscapes. How easy is to determine the automorphism of a genotype network. 
